# EXGC: Bridging Efficiency and Explainability in Graph Condensation

Anonymity

## ABSTRACT

Graph representation learning on vast datasets, like web data, has made significant strides. However, the associated computational and storage overheads raise concerns. In sight of this, Graph condensation (GCond) has been introduced to distill these large real datasets into a more concise yet information-rich synthetic graph. Despite acceleration efforts, existing GCond methods mainly grapple with efficiency, especially on expansive web data graphs. Hence, in this work, we pinpoint two major inefficiencies of current paradigms: (1) the concurrent updating of a vast parameter set, and (2) pronounced parameter redundancy. To counteract these two limitations correspondingly, we first (1) employ the Mean-Field variational approximation for convergence acceleration, and then (2) propose the objective of Gradient Information Bottleneck (GDIB) to prune redundancy. By incorporating the leading explanation techniques (*e.g.*, GNNExplainer and GSAT) to instantiate the GDIB, our EXGC, the **E**fficient and e**X**plainable **G**raph **C**ondensation method is proposed, which can markedly boost efficiency and inject explainability. Our extensive evaluations across eight datasets underscore EXGC's superiority and relevance. Code is available at https://anonymous.4open.science/r/EXGC.

## KEYWORDS

Graph Neural Networks, Graph Condensation, Model Explainability

## 1 INTRODUCTION

Web data, such as social networks [12], transportation systems [45, 70], and recommendation platforms [51, 52], are often represented as graphs. These graph structures are ubiquitous in everyday activities, including streaming on Netflix, interacting on Facebook, shopping on Amazon, or searching on Google [2, 68]. Given their tailor-made designs, Graph Neural Networks (GNNs) [10, 19, 25] have emerged as a prevalent solution for various tasks on graph-structured data and showcased outstanding achievements across a broad spectrum of graph-related web applications [18, 22, 62, 69].

However, real-world scenarios often entail the handling of large-scale graphs encompassing millions of nodes and edges [21, 28], posing substantial computational burdens during the training of GNN applications [9, 56, 60]. Worse still, the challenges are exacerbated when fine-tuning hyperparameters and discerning optimal training paradigms for over-parametrized GNN models. Against this backdrop, a crucial inquiry arises: *can we effectively simplify or reduce the graph size to accelerate graph algorithm operations, including GNNs, while also streamlining storage, visualization, and retrieval essential for graph data analysis [23, 42, 64]?*

As a primary solution, graph sampling emphasizes selecting pivotal edges/nodes and omitting the less relevant ones [5, 6, 11, 41]. However, this can lead to considerable information loss, potentially harming model performance [44, 50]. Conversely, graph distillation aims to compress the extensive real graph $\mathcal{T}$ into a concise yet

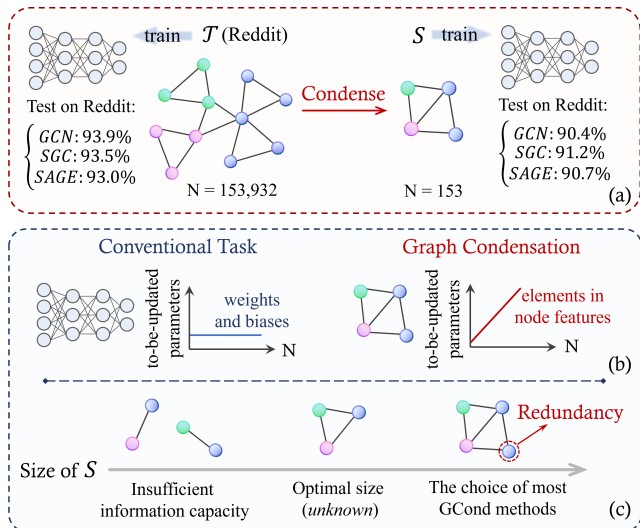

**Figure 1: The compression capability and limitations of current GCond. (a) GCond adeptly compresses the dataset to just 0.1% of its initial size without compromising the accuracy benchmarks. (b) Contrary to traditional graph learning, GCond's parameters scale with node count. (c) To avoid insufficient information capacity, GCond typically introduces node redundancy.**

information-rich synthetic graph $\mathcal{S}$, enhancing the efficiency of the graph learning training process. Within this domain, the **graph condensation** (GCond) stands out due to its exceptional compression capabilities [23, 24]. For instance, as depicted in Figure 1 (a), the graph learning model trained on the synthetic graph $\mathcal{S}$ (containing just 154 nodes generated by GCond) yields a 91.2% test accuracy on Reddit, nearly matching the performance of the model trained on the original dataset with 153,932 nodes (*i.e.*, 93.5% accuracy).

Despite their successes, we argue that even with various acceleration strategies, current GCond methods remain facing efficiency challenges in the training process, particularly on large graph datasets such as web data. This inefficiency arises from two main factors:

- Firstly, as depicted in Figure 1 (b), the **prolonged convergence** stems from the concurrent updating of an overwhelming number of parameters (*i.e.*, elements in node features of $\mathcal{S}$). Specifically, unlike conventional graph learning where parameter dimensionality is dataset-agnostic, in GCond, the number of parameters grows with the nodes and node feature dimensions, imposing substantial computational and storage demands.
- Secondly, as illustrated in Figure 1 (c), the current GCond approaches mainly exhibit **node redundancy**. Concretely, when compressing new datasets, to counteract the risk of insufficient information capacity from too few nodes, a higher node count

is typically employed by $\mathcal{S}$, leading to parameter redundancy in the training process. Depending on the dataset attributes, this redundancy can vary, with some instances exhibiting as much as 92.7% redundancy. We put further discussion in Section 3.3.

In sight of this, in this work, we aim to refine the paradigm of GCond to mitigate the above limitations. Specifically, **for the first limitation**, we scrutinize and unify the paradigms of the current methods from the perspective of Expectation Maximization (EM) framework [8, 33], and further formulate it as the theoretical basis for our forthcoming optimization schema. From this foundation, we pinpoint the efficiency bottleneck in the training process, *i.e.*, the computation of intricate posterior probabilities during the Expectation step (E-step). This insight led us to employ the Mean-Field (MF) variational approximation [3] – a renowned technique for improving the efficiency of E-step with intricate variables – to revise the paradigm of GCond. The streamlined method is termed Mean-Field Graph Condensation (MGCond).

Then, **for the second limitation**, our solution seeks to 'explain' the training process of the synthetic graph $\mathcal{S}$: we prioritize the most informative nodes in $\mathcal{S}$ (*i.e.*, nodes encapsulating essential information for model training) and exclude the remaining redundant nodes from the training process. To formulate this objective, inspired by the principle of graph information bottleneck, we introduce the Gradient Information Bottleneck (GDIB). Building upon GDIB, our EXGC, the **E**fficient and e**X**plainable **G**raph **C**ondensation method, is proposed by integrating the leading explanation strategies (*e.g.*, GNNExplainer [57] and GSAT [32]) into the paradigm of MGCond.

Our contribution can be summarized as follow:

- For the limitation of inefficiency, we unify the paradigms of current approaches to pinpoint the cause and leverage Mean-Field variational approximation to propose the MGCond for boosting efficiency (Section 3.1 & 3.2).
- For the caveat posed by node redundancy, we introduce the objective of Gradient Information Bottleneck, and utilize the leading explanation methods to develop an explainable and efficient method, EXGC (Section 3.3 & 3.4).
- Extensive experiments demonstrate that our EXGC outperforms the baselines by a large margin. For instance, EXGC is 11.3 times faster than the baselines on Citeseer (Section 4).

Furthermore, it is worth mentioning that beyond the tasks of graph condensation, the superior performance of EXGC across various backbones (*i.e.*, explainers) also verifies the effectiveness of the graph explanation methods in enhancing downstream graph tasks. To our knowledge, this stands as one of the vanguard efforts in the application of graph explainability, addressing a crucial yet rarely explored niche.

## 2 PROBLEM FORMULATION

In this part, we retrospect the objective of graph condensation. Specifically, graph condensation endeavors to transmute a large, original graph into a compact, synthetic, and highly informative counterpart. The crux of this process is to ensure that the GNNs trained on the condensed graph manifest a performance comparable to those trained on the original graph.

**Notations.** Initially, we delineate the common variables utilized in this study. We start from the original graph $\mathcal{T} = (\mathbf{A}, \mathbf{X}, \mathbf{Y})$, where $\mathbf{A} \in \mathbb{R}^{N \times N}$ is the adjacency matrix, $N$ is the number of nodes and $\mathbf{X} \in \mathbb{R}^{N \times d}$ is the $d$-dimensional node feature attributes. Further, we note the label of nodes as $\mathbf{Y} = \{0, 1, \ldots, C - 1\}^N$ denotes the node labels over $C$ classes. Our target is to train a synthetic graph $\mathcal{S} = (\mathbf{A}', \mathbf{X}', \mathbf{Y}')$ with adjacency matrix $\mathbf{A}' \in \mathbb{R}^{N' \times N'}$ and feature attributes $\mathbf{X}' \in \mathbb{R}^{N' \times D}$ ($N' \ll N$), which can achieve comparable performance with $\mathcal{T}$ under GNNs inference process.

**Graph condensation via gradient matching.** The above objective of graph condensation can be formulated as follows:

$$\min_{\mathcal{S}} \mathcal{L}\left(f_{\theta_{\mathcal{S}}}(\mathbf{A}, \mathbf{X}), \mathbf{Y}\right),$$
$$\text{s.t.} \ \ \theta_{\mathcal{S}} = \arg\min_{\theta} \mathcal{L}\left(f_{\theta}(\mathbf{A}', \mathbf{X}'), \mathbf{Y}'\right), \tag{1}$$

where $\mathcal{L}$ represents the loss function and $f_{\theta}$ denotes the graph learning model $f$ with parameters $\theta$. In pursuit of this objective, the previous works typically employ the gradient matching scheme following [23, 24, 66]. Concretely, given a graph learning model $f_{\theta}$, these methods endeavor to reduce the difference of model gradients *w.r.t.* real data $\mathcal{T}$ and synthetic data $\mathcal{S}$ for model parameters [23]. Hence, the graph learning models trained on synthetic data will converge to similar states and share similar test performance with those trained on real data.

## 3 METHODOLOGY

In this section, we first unify the paradigms of current GCond methods in Section 3.1. Building upon this, we propose the MGCond, which employs MF approximation to boost efficiency in Section 3.2. Furthermore, to eliminate the redundancy in the training process, we introduce the principle of GDIB in Section 3.3 and instantiate it to develop our EXGC in Section 3.4.

### 3.1 The Unified Paradigm of GCond

As depicted in Section 2, graph condensation aims to match the model gradients *w.r.t* large-real graph $\mathcal{T}$ and small-synthetic graph $\mathcal{S}$ for model parameters. This process enables GNNs trained on $\mathcal{T}$ and $\mathcal{S}$ to share a similar training trajectory and ultimately converge to similar states (parameters). We formulate this gradient matching process as follows:

$$\max_{\mathcal{S}} \mathrm{E}_{\theta \sim \mathbb{P}_{\theta}} P(\nabla'_{\theta} = \nabla_{\theta}),$$
$$\text{s.t.} \ \nabla'_{\theta} = \frac{\partial \mathcal{L}(f_{\theta}(\mathcal{S}), \mathbf{Y}')}{\partial_{\theta}}, \nabla_{\theta} = \frac{\partial \mathcal{L}(f_{\theta}(\mathcal{T}), \mathbf{Y})}{\partial_{\theta}}, \tag{2}$$

where $\mathbb{P}_{\theta}$ denotes the distribution of $\theta$'s potential states during the training process. For example, [24] defines $\mathbb{P}_{\theta}$ as the set of parameter states that can appear throughout a complete network training process, while [23] simply defines it as the potential initial states of the parameters.

Considering the computational complexity of jointly optimizing $\mathbf{X}'$, $\mathbf{A}'$, and $\mathbf{Y}'$, and the interdependency between these three variables, current methods typically fix the labels $\mathbf{Y}'$ and design a MLP-based model $g_{\Phi}$ with parameters $\Phi$ to calculate $\mathbf{A}'$ following

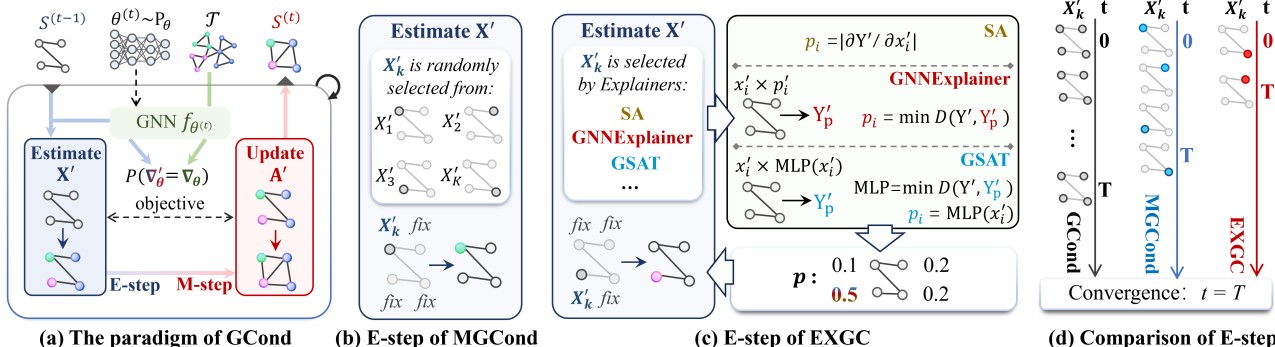

**Figure 2: The paradigm of current GCond methods from the perspective of the EM schema, and the E-step of our proposed MGcond and EXGC.**

$A' = g_\Phi(X')$ [24]. In this case, Equation 2 can be rewrite as:

$$\max_{X', \Phi} E_{\theta \sim \mathbb{P}_\theta} P(\nabla'_\theta = \nabla_\theta),$$

$$\text{s.t. } \nabla'_\theta = \frac{\partial \mathcal{L}(f_\theta(X', g_\Phi(X')), Y')}{\partial \theta}, \nabla_\theta = \frac{\partial \mathcal{L}(f_\theta(X, A), Y)}{\partial \theta}. \quad (3)$$

Without loss of generality, $\nabla'_\theta$ and $\nabla_\theta$ are consistently defined as provided here in the following text, even though not all previous methods have employed the MLP-based simplification strategy [1].

After random initialization, Equation 3 can be achieved by alternately optimizing the variable $X'$ and the model parameters $\Phi$, which naturally adheres to the Expectation-Maximization schema, as shown in Figure 2 (a). Specifically, the EM algorithm alternates between the expectation step (E-step) and the maximization step (M-step):

- **E-step:** Estimate the variable $X'$ while freezing the model $g_\Phi$, then utilize it to calculate the Evidence Lower Bound (ELBO) of the objective of the gradient matching in Equation 3.
- **M-step:** Fine the parameters $\Phi$ which maximizes the above ELBO.

After instantiating the above schema, graph condensation can be formulated as follows, where $t$ represents the training epoch and $\nabla_\theta$ is a simplified notation for $\nabla'_\theta = \nabla_\theta$:

- **Initialization:** Select the initial value of the parameter $\Phi^{(0)}$ and the node feature $X'^{(0)}$, then start the iteration;
- **E-step:** Use the model $g(\Phi^{(t)})$ to estimate the node features $X'^{(t)}$ according to $P(X'^{(t)}|\nabla_\theta, \Phi^{(t)})$ and calculate the ELBO:

$$\text{ELBO} \rightarrow E_{X'^{(t)}|\nabla_\theta, \Phi^{(t)}}[\log \frac{P(X'^{(t)}, \nabla_\theta \mid \Phi)}{P(X'^{(t)} \mid \nabla_\theta, \Phi^{(t)})}]; \quad (4)$$

- **M-step:** Find the corresponding parameters $\Phi^{(t+1)}$ when the above ELBO is maximized:

$$\Phi^{(t+1)} := \arg\max_\Phi E_{X'^{(t)}|\nabla_\theta, \Phi^{(t)}}[\log \frac{P(X'^{(t)}, \nabla_\theta \mid \Phi)}{p(X'^{(t)} \mid \nabla_\theta, \Phi^{(t)})}]; \quad (5)$$

- **Output:** Repeat the E-step and M-step until convergence, then output the synthetic graph $\mathcal{S}$ according to the final $X'$ and $\Phi$.

The detailed derivation of the above Equations is shown in Appendix B.

---

[1]For methods that do not adopt the simplification strategy, by replacing $g_\Phi$ with $A$, the subsequent theoretical sections still hold true.

**Revealing the Limitation of Inefficiency.** However, we have noticed that even with various acceleration strategies [24], the above paradigm remains facing efficiency challenges in the training process. We attribute this limitation to the estimation process of $X'$ in E-step. Specifically, in contrast to traditional graph learning tasks where the number of network parameters is **dataset-agnostic**, for graph condensation task, the number of to-be-updated parameters in E-step (*i.e.*, elements in $X'$) linearly **increases** with the number of nodes $N$ and feature dimensions $d$, posing substantial burden of gradient computation and storage.

This flaw is particularly evident on large graph datasets such as web data with millions of nodes ($N$) and thousands of feature dimensions ($d$). Therefore, it is crucial to find a shortcut for expediting the current paradigm.

## 3.2 Boost Efficiency: MGCond

To address the limitation of inefficiency, we aim to inject the Mean-Field variational approximation [3] into the current GCond paradigm. In practice, MF approximation has been extensively verified to enhance the efficiency of the EM framework containing variables with complex distributions. Hence, it precisely matches the challenge encountered in our E-step, where the to-be-updated variable $X'$ possesses large dimensions. Next, we elucidate the process of leveraging MF estimation to enhance the GCond paradigm.

Firstly, MF approximation assumes that the to-be-updated variable can be decomposed into multiple independent variables, aligning naturally with the property of node features $X' = \{x'_1, x'_2, ..., x'_{N'}\}$ of $\mathcal{S}$ in our E-step (*i.e.*, Equation 4):

$$P(X') = \prod_{i=1}^{N'} P(x'_i), \quad (6)$$

where $x'_i$ is the feature of the $i$-th node in graph $\mathcal{S}$. By substituting Equation 6 into the ELBO in Equation 4 we obtain:

$$\text{ELBO} = \int \prod_{i=1}^{N'} P(x'_i) \log P(\nabla_\theta, X') dX' $$
$$- \int \prod_{i=1}^{N'} P(x'_i) \log \prod_{i=1}^{N'} P(x'_i) dX'. \quad (7)$$

In this case, while we focus on the node feature $x'_j$ and fix its complementary set $\mathbf{X}'_{\backslash j} = \{x'_1, ...x'_{j-1}, x'_{j+1}, ..., x'_{N'}\}$, the ELBO in Equation 7 can be rewritten as:

$$\text{ELBO} = \int P\left(x'_j\right) \int \prod_{i=1, i\neq j}^{N'} P\left(x'_i\right) \log P(\nabla_\theta, \mathbf{X}') d_{i\neq j} x'_i dx'_j$$

$$- \int P\left(x'_j\right) \log P\left(x'_j\right) dx'_j + \sum_{i=1, i\neq j}^{N'} \int P\left(x'_i\right) \log P\left(x'_i\right) dx'_i, \quad (8)$$

where the third term can be considered as the constant $C$ because $\mathbf{X}'_{\backslash j}$ is fixed. Then, to simplify the description, we define:

$$\log \tilde{P}_j(\mathbf{X}', \nabla_\theta) = E_{\prod_{i=1, i\neq j}^{N'} P(x'_i)}[\log P(\mathbf{X}', \nabla_\theta)]$$

$$= \int \prod_{i=1, i\neq j}^{N'} P\left(x'_i\right) \log P(\mathbf{X}', \nabla_\theta) d_{i\neq j} x'_i, \quad (9)$$

and combine it with Equation 8 to obtain the final form of the ELBO which is streamlined by the MF variational approximation:

$$\text{ELBO} = \int P(x'_j) \frac{\log \tilde{P}_j(\mathbf{X}', \nabla_\theta)}{P(x'_j)} dx'_j + C$$

$$= -KL\left(P(x'_j) \| \log \tilde{P}(\mathbf{X}', \nabla_\theta)\right) + C, \quad (10)$$

where $KL$ denotes the Kullback-Leibler (KL) Divergence [7]. Due to the non-negativity of the KL divergence, maximizing this ELBO is equivalent to equating the two terms in the above KL divergence. Based on this, we have:

$$P(\mathbf{X}') \propto \prod_{j=1}^{N'} \log \tilde{P}_j(\mathbf{X}', \nabla_\theta), \quad (11)$$

which can be regarded as the theoretical guidance for the $\mathbf{X}'$ estimation process in the E-step. The detailed derivation is exhibited in Appendix C.

**The Paradigm of MGCond.** Equation 11 indicates that the estimation of node feature $x'_j$ in E-step can be performed while keeping its complementary features $\mathbf{X}'_{\backslash j}$ fixed. Without loss of generality, we generalize this conclusion from individual nodes to subsets of nodes, and distribute the optimization process of each set evenly over multiple iterations. This optimized E-step is the key distinction between our MGCond and the prevailing paradigm, as illustrated in Figure 2 (b). To be more specific, the paradigm of MGCond can be formulated as follows:

- **Initialization:** Select the initial value of the parameter $\Phi^{(0)}$ and features $\mathbf{X}'^{(0)}$, divide the nodes in graph $\mathcal{S}$ into $K$ parts equally i.e., $\mathbf{X}' = \{\mathbf{X}'_1, \mathbf{X}'_2, ..., \mathbf{X}'_K\}$, and start the iteration;
- **E-step:** Use the model $g(\Phi^{(t)})$ to estimate the features in subsets $\mathbf{X}'^{(t)}_k$ for $k = \{1, 2, ..., K\}$ according to:

$$\mathbf{X}'^{(t+1)}_k = \begin{cases} \max_{\mathbf{X}'_k} P(\mathbf{X}'_k | \mathbf{X}'^{(t)}_{\backslash k}, \nabla_\theta, \Phi^{(t)}), & \text{if } k = r + 1, \\ \mathbf{X}'^{(t)}_k, & \text{otherwise,} \end{cases} \quad (12)$$

where $r$ is the remainder when $t$ is divided by $K$.

- **M-step:** Find the corresponding parameters $\Phi^{(t+1)}$ when the following ELBO is maximized:

$$\Phi^{(t+1)} := \arg\max_\Phi E_{\mathbf{X}'^{(t+1)} | \nabla_\theta, \Phi^{(t)}}[\log \frac{P(\mathbf{X}'^{(t+1)}, \nabla_\theta \mid \Phi)}{p(\mathbf{X}'^{(t+1)} \mid \nabla_\theta, \Phi^{(t)})}]; \quad (13)$$

- **Output:** Repeat the E-step and M-step until convergence, then output the synthetic graph $\mathcal{S}$ according to the final $\mathbf{X}'$ and $\Phi$.

### 3.3 Node Redundancy and GDIB

After executing MGCond we summarize two empirical insights that primarily motivated the development of our XEGC as follows:

(1) The training process of $\mathbf{X}'$ in E-step exhibits a *long-tail problem*. That is, when **20%** of the node features $\mathbf{X}'$ are covered in training (*i.e.*, $t \approx 0.2K$), the improvement in test accuracy has already achieved **93.7%** of the total improvement on average. In other words, the remaining 80% of the node features only contribute to 6.3% of the accuracy improvement.

(2) This long-tail problem has a larger variance. Specifically, even for the same task with the same setting and initialization, when 20% of the $\mathbf{X}'$ are covered in training, the maximum difference between test accuracy exceeds 25% (*i.e.*, difference between 72.4% and 98.8%), since those 20% trained nodes are randomly selected from $\mathcal{S}$.

These two observations indicate that there is a considerable **redundancy** in the number of to-be-trained nodes. That is, the synthetic graph $\mathcal{S}$ comprises a subset of key nodes that possess most of the necessary information for gradient matching. If the initial random selections pinpoint these key nodes, the algorithm can yield remarkably high test accuracy in the early iterations. On the other side, entirely training all node features $\mathbf{X}'$ in $\mathcal{S}$ would not only be computationally wasteful but also entail the potential risk of overfitting the given graph learning model.

Therefore, it naturally motivates us to identify and train these key nodes in E-step (instead of randomly selecting nodes to participate in training like MGCond). To guide this process, inspired by the Graph Information Bottleneck (GIB) for capturing key subgraphs [53, 61] and guiding GNNs explainability [13, 32], we propose the GraDient Information Bottleneck (GDIB) for the compact graph condensation with the capability of redundancy removal:

**Definition 1 (GDIB):** *Given the the synthetic graph $\mathcal{S}$ with label $\mathbf{Y}'$ and the GNN model $f_\theta$, GDIB seeks for a maximally informative yet compact subgraph $\mathcal{S}_{sub}$ by optimizing the following objective:*

$$\arg\max_{\mathcal{S}_{sub}} I\left(\mathcal{S}_{sub}; \nabla'_\theta\right) - \beta I\left(\mathcal{S}_{sub}; \mathcal{S}\right), \text{ s.t. } \mathcal{S}_{sub} \in \mathbb{G}_{sub}(\mathcal{S}), \quad (14)$$

*where $\nabla'_\theta$ denotes the gradients $\partial\mathcal{L}(f_\theta(\mathcal{S}), \mathbf{Y}')/\partial\theta$; $\mathbb{G}_{sub}(\mathcal{S})$ indicates the set of all subgraphs of $\mathcal{S}$; $I$ represents the mutual information (MI) and $\beta$ is the Lagrangian multiplier.*

### 3.4 Prune Redundancy: EXGC

**A Tractable Objective of GDIB.** To pinpoint the crucial node features to participate in the training process in E-step, we first derive a tractable variational lower bound of the GDIB. Detailed derivation can be found in Appendix D, which is partly adapted from [32, 61].

Specifically, for the first term $I(\mathcal{S}_{sub}; \nabla'_\theta)$, a parameterized variational approximation $Q(\nabla'_\theta|\mathcal{S}_{sub})$ for $P(\nabla'_\theta|\mathcal{S}_{sub})$ is introduced to derive its lower bound:

$$I(\mathcal{S}_{sub}; \nabla'_\theta) \geq \mathbb{E}_{\mathcal{S}_{sub};\nabla'_\theta}\left[\log Q(\nabla'_\theta|\mathcal{S}_{sub})\right]. \quad (15)$$

For the second term $I(\mathcal{S}_{sub}; \mathcal{S})$, we introduce the variational approximation $R(\mathcal{S}_{sub})$ for the marginal distribution $P(\mathcal{S}_{sub}) = \sum_{\mathcal{S}} R(\mathcal{S}_{sub}|\mathcal{S}) P(\mathcal{S})$ to obtain its upper bound:

$$I(\mathcal{S}_{sub}; \mathcal{S}) \leq \mathbb{E}_{\mathcal{S}}\left[\text{KL}\left(P(\mathcal{S}_{sub}|\mathcal{S}) \| R(\mathcal{S}_{sub})\right)\right]. \quad (16)$$

By incorporating the above two inequalities, we derive a variational upper bound for Equation 14, serving as the objective for

$$\underset{\mathcal{S}_{sub}}{\arg\max} \mathbb{E}\left[\log Q(\nabla'_\theta|\mathcal{S}_{sub})\right] - \mathbb{E}\left[\text{KL}\left(P(\mathcal{S}_{sub}|\mathcal{S}) \| R(\mathcal{S}_{sub})\right)\right]. \quad (17)$$

**Instantiation of the GDIB.** To achieve the above upper bound, we simply adopt $\partial\mathcal{L}(f_\theta(\mathcal{S}_{sub}), \mathbf{Y}')/\partial\theta$ to instantiate the distribution $Q$. Then, we specify the distribution $R$ in Equation 16 as a Bernoulli distribution with parameter $r$ (*i.e.*, each node is selected with probability $r$). As for $P(\mathcal{S}_{sub}|\mathcal{S})$, we suppose it assigns the importance score $p_i$ (*i.e.*, the probability of being selected into $\mathcal{S}_{sub}$) to the $i$-th node in $\mathcal{S}$. After that, GDIB can be instantiated by the post-hoc explanation methods such as:

- **Gradient-based** methods like SA [1] and GradCAM [40]. For the $i$-th node, these methods first calculate the absolute values of the elements in the derivative of $\mathcal{L}(f_\theta(\mathcal{S}), \mathbf{Y}')$ *w.r.t* $x_i$ (*i.e.*, the features of the $i$-th node). After that, the importance score $p_i$ is defined as the normalized sum of these values. More formally:

$$p_i = \underset{i\in[1,2,...,N]}{\text{softmax}}\left(|\frac{\partial\mathcal{L}(f_\theta(\mathcal{S}), \mathbf{Y}')}{\partial x_i}| \cdot \mathbf{1}^\mathbf{T}\right). \quad (18)$$

- **Local Mask-based** methods like GNNExplainer [57] and Graph-MASK [39]. Concretely, for the first term of Equation 17, these methods firstly multiply the node's features $x'_i$ with the initialized node importance score $p_i$ to get $\mathbf{X}'' = \{p_1 x'_1, p_2 x'_2, ..., p_{N'} x'_{N'}\}$, and feed $\mathbf{X}''$ into model $f_\theta$ to obtain the output $y_p$. Then they attempt to find the optimal score $p_i$ by minimizing the difference between this processed output $y$ and the original prediction. Concurrently, for the second term of Equation 17, these methods set $r$ to approach 0, making the value of the KL divergence proportional to the score $p_i$. As a result, they treat this KL divergence as the $l_1$-norm regularization term acting on $p_i$ to optimize the training process of $p_i$. After establishing these configurations, the optimal score can be approximated through several gradient descents following:

$$\mathbf{p} = \min_\mathbf{p} D\left(y; y_p\right) + \lambda\mathbf{p}\cdot\mathbf{1}^\mathbf{T}, \quad (19)$$

where $\mathbf{p}$ is defined as $\{p_1, p_2, ..., p_N\}$; $D$ denotes the distance function; $\lambda$ is the trade-off parameter; $y$ and $y_p$ represents:

$$\begin{cases} y = f_\theta(\{\mathbf{X}', g_\Phi(\mathbf{X}')\}), \\ y_p = f_\theta(\{\mathbf{X}'', g_\Phi(\mathbf{X}'')\}), \end{cases} \quad (20)$$

- **Global Mask-based** methods like GSAT[2] [32] and PGExplainer [31]. Here, during the instantiation process of the first term of Equation 17, the trainable $p_i$ in Local Mask-based methods is

replaced with a trainable MLP$_\psi$ (*i.e.*, $p_i = \text{MLP}_\psi(x'_i)$) and $y_p$ is correspondingly replaced with $y_{\text{MLP}}$. Meanwhile, for the second term in Equation 17, these methods set $r \in (0, 1)$ to instantiate the KL divergence as the *information constraint* ($\ell_I$) proposed by [32], where $\ell_I$ is defined as:

$$\ell_I = \sum_{i\in1,2,...,N} p_i \log\frac{p_i}{r} + (1 - p_i)\log\frac{1 - p_i}{1 - r}. \quad (21)$$

Treating $\ell_I$ as a regularization term acting on $\mathbf{p}$, the explainers can obtain the approximate optimal score $\mathbf{p}$ through several gradient optimizations of $\psi$ following:

$$\psi = \min_\psi D\left(y; y_{\text{MLP}}\right) + \lambda\ell_I, \quad (22)$$

After obtaining the importance score $p_i$, the crucial subgraph $\mathcal{S}_{sub}$ in GDIB can be composed of nodes with larger scores $p_i$.

**The Paradigm of EXGC.** As illustrated in Figure 2 (c), after leveraging the above leading post-hoc graph explanation methods to achieve the objective of GDIB, we summarize the paradigm of our EXGC as follows:

- **Initialization:** Select the initial value of the parameter $\Phi^{(0)}$, the node features $\mathbf{X}'^{(0)}$, the set of the node index $\mathbf{M}$ and the ratio of nodes optimized in each E-step as $\kappa$, then start the iteration;

- **E-step:** Leverage the above explainers to assign an importance score $p_i$ to the $i$-th node in $\mathcal{S}$ for the index $i$ in set $\mathbf{M}$:

$$\{p_i\} = \text{Explainer}\left(\{x_i\}, f_\theta\right), \quad \text{for } i \in \mathbf{M}. \quad (23)$$

Subsequently, remove the indices corresponding to the nodes with the top $\lfloor\kappa N'\rfloor$ scores from set $\mathbf{M}$. Then use the model $g(\Phi^{(t)})$ to estimate the features $\mathbf{X}'^{(t+1)}$ according to:

$$\begin{cases} \mathbf{X}'^{(t+1)}_\mathbf{M} = \max_{\mathbf{X}'_\mathbf{M}} P(\mathbf{X}'_\mathbf{M}|\mathbf{X}'^{(t)}_{\backslash\mathbf{M}}, \nabla_\theta, \Phi^{(t)}), \\ \mathbf{X}'^{(t+1)}_{\backslash\mathbf{M}} = \mathbf{X}'^{(t)}_{\backslash\mathbf{M}}, \end{cases} \quad (24)$$

where $\mathbf{X}'_\mathbf{M} = \{x_i\}$ for $i \in \mathbf{M}$, and $\mathbf{X}'_{\backslash\mathbf{M}} = \mathbf{X}' \backslash \mathbf{X}'_\mathbf{M}$.

- **M-step:** Find the corresponding parameters $\Phi^{(t+1)}$ when the following ELBO is maximized:

$$\Phi^{(t+1)} := \underset{\Phi}{\arg\max} E_{\mathbf{X}'^{(t+1)}|\nabla_\theta,\Phi^{(t)}}\left[\log\frac{P(\mathbf{X}'^{(t+1)}, \nabla_\theta \mid \Phi)}{p(\mathbf{X}'^{(t+1)} \mid \nabla_\theta, \Phi^{(t)})}\right]; \quad (25)$$

- **Output:** Repeat the E-step and M-step until convergence, then output the synthetic graph $\mathcal{S}$ according to the final $\mathbf{X}'$ and $\Phi$.

The comparison between E-steps in the paradigms of GCond, MGCond and EXGC is exhibited in Figure 2 (d). By leveraging graph explanation methods to instantiate the objective of GDIB and seamlessly integrating it within the MGCond's training paradigm, our proposed EXGC adeptly identifies pivotal nodes in the synthetic graph $\mathcal{S}$ during early training stages. Experimental results in the ensuing section underline that EXGC frequently converges early – specifically when a mere **20%** of the nodes in $\mathcal{S}$ participate in training – attributed to the successful identification of these key nodes. EXGC's computational focus on these essential nodes ensures resource optimization, precluding superfluous expenditure on extraneous nodes. As a result, it can not only boost the efficiency but also enhance the test accuracy.

---

[2]The GSAT mentioned here refers to the GSAT in the post-explanation mode [32].

**Table 1: Test performance (%) comparison among EXGC and other baselines, from which we can easily find that EXGC achieves promising performance in comparison to baselines even with extremely large reduction rates. $\rho$ denotes the inference speedup. In this table, we only display the EXGC based on Global Mask-based Explainers.**

| Dataset | Ratio | Baselines | | | Ablation | | Ours | | | Storage | $\rho$ |
|---|---|---|---|---|---|---|---|---|---|---|---|
| | | Random | Herding | K-Center | GCond-X | GCond | EXGC-X | EXGC | Full graph | | |
| Citeseer (47.1M) | 0.3% | $33.87_{\pm0.82}$ | $31.31_{\pm1.20}$ | $34.03_{\pm2.52}$ | $64.13_{\pm1.83}$ | $63.98_{\pm4.31}$ | $67.82_{\pm1.31}$ | $69.16_{\pm2.00}$ | $71.12_{\pm0.06}$ | 0.142M | 333.3× |
| Citeseer (47.1M) | 1.8% | $42.66_{\pm1.30}$ | $40.61_{\pm2.13}$ | $51.79_{\pm3.24}$ | $67.24_{\pm1.85}$ | $66.82_{\pm2.70}$ | $69.60_{\pm1.88}$ | $70.09_{\pm0.72}$ | $71.12_{\pm0.06}$ | 0.848M | 55.6× |
| Citeseer (47.1M) | 3.6% | $59.74_{\pm2.85}$ | $63.85_{\pm1.77}$ | $67.25_{\pm1.60}$ | $69.86_{\pm0.97}$ | $69.74_{\pm1.36}$ | $70.18_{\pm1.17}$ | $70.55_{\pm0.93}$ | $71.12_{\pm0.06}$ | 1.696M | 27.8× |
| Cora (14.9M) | 0.4% | $37.04_{\pm7.41}$ | $43.47_{\pm0.55}$ | $46.33_{\pm3.24}$ | $69.10_{\pm0.31}$ | $72.76_{\pm0.45}$ | $80.91_{\pm0.39}$ | $82.02_{\pm0.42}$ | $80.91_{\pm0.10}$ | 0.060M | 250.0× |
| Cora (14.9M) | 1.3% | $59.62_{\pm2.48}$ | $62.18_{\pm1.91}$ | $69.12_{\pm2.55}$ | $75.38_{\pm1.59}$ | $79.29_{\pm0.76}$ | $80.74_{\pm0.41}$ | $81.94_{\pm1.03}$ | $80.91_{\pm0.10}$ | 0.194M | 76.9× |
| Cora (14.9M) | 2.6% | $73.29_{\pm1.03}$ | $70.91_{\pm2.12}$ | $73.66_{\pm1.85}$ | $75.98_{\pm0.93}$ | $80.02_{\pm0.69}$ | $81.65_{\pm0.77}$ | $82.26_{\pm0.90}$ | $80.91_{\pm0.10}$ | 0.388M | 38.5× |
| Ogbn-arxiv (100.4M) | 0.05% | $46.83_{\pm2.60}$ | $49.74_{\pm2.30}$ | $47.28_{\pm1.15}$ | $56.49_{\pm1.69}$ | $57.39_{\pm0.65}$ | $58.46_{\pm0.85}$ | $57.62_{\pm0.64}$ | $70.76_{\pm0.04}$ | 0.050M | 2000.0 × |
| Ogbn-arxiv (100.4M) | 0.25% | $57.32_{\pm1.19}$ | $58.64_{\pm1.28}$ | $54.36_{\pm0.67}$ | $62.38_{\pm1.62}$ | $62.49_{\pm1.56}$ | $64.82_{\pm0.51}$ | $62.34_{\pm0.26}$ | $70.76_{\pm0.04}$ | 0.251M | 400.0× |
| Ogbn-arxiv (100.4M) | 0.5% | $60.09_{\pm0.97}$ | $61.25_{\pm0.88}$ | $60.84_{\pm0.59}$ | $63.77_{\pm0.95}$ | $64.85_{\pm0.74}$ | $65.79_{\pm0.32}$ | $64.99_{\pm0.79}$ | $70.76_{\pm0.04}$ | 0.502M | 200× |
| Ogbn-Product (1412.5M) | 0.5% | $57.49_{\pm2.53}$ | $60.10_{\pm0.36}$ | $59.46_{\pm1.22}$ | $61.59_{\pm0.61}$ | $62.15_{\pm0.36}$ | $62.71_{\pm0.91}$ | $62.09_{\pm0.74}$ | $70.76_{\pm0.04}$ | 7.063M | 200.0× |
| Ogbn-Product (1412.5M) | 1.5% | $58.84_{\pm1.87}$ | $63.17_{\pm0.93}$ | $60.71_{\pm0.85}$ | $62.98_{\pm1.30}$ | $63.89_{\pm0.51}$ | $65.85_{\pm0.95}$ | $64.69_{\pm1.43}$ | $70.76_{\pm0.04}$ | 21.189M | 66.7× |
| Ogbn-Product (1412.5M) | 3% | $60.19_{\pm0.47}$ | $63.87_{\pm0.41}$ | $62.60_{\pm1.38}$ | $65.82_{\pm0.59}$ | $65.30_{\pm0.92}$ | $67.50_{\pm1.05}$ | $66.37_{\pm0.72}$ | $70.76_{\pm0.04}$ | 42.378M | 33.3× |
| Flickr (86.8M) | 0.1% | $41.84_{\pm1.87}$ | $43.90_{\pm0.56}$ | $43.30_{\pm0.90}$ | $46.93_{\pm0.10}$ | $46.81_{\pm0.10}$ | $46.95_{\pm0.03}$ | $47.01_{\pm0.10}$ | $47.16_{\pm0.17}$ | 0.087M | 1000.0× |
| Flickr (86.8M) | 0.5% | $44.64_{\pm0.52}$ | $43.95_{\pm44.17}$ | $44.17_{\pm0.33}$ | $45.91_{\pm0.08}$ | $46.97_{\pm1.14}$ | $47.83_{\pm0.95}$ | $48.29_{\pm0.45}$ | $47.16_{\pm0.17}$ | 0.434M | 200.0× |
| Flickr (86.8M) | 1% | $44.89_{\pm1.25}$ | $44.67_{\pm0.57}$ | $44.68_{\pm0.69}$ | $45.72_{\pm0.71}$ | $47.01_{\pm0.65}$ | $47.62_{\pm0.10}$ | $48.36_{\pm0.88}$ | $47.16_{\pm0.17}$ | 0.868M | 100.0× |
| Reddit (435.5M) | 0.1% | $59.14_{\pm2.26}$ | $65.75_{\pm1.28}$ | $53.05_{\pm2.73}$ | $89.34_{\pm0.54}$ | $89.56_{\pm0.74}$ | $89.56_{\pm0.45}$ | $90.24_{\pm0.05}$ | $93.96_{\pm0.03}$ | 0.436M | 1000.0× |
| Reddit (435.5M) | 0.2% | $65.38_{\pm2.68}$ | $71.92_{\pm1.17}$ | $58.64_{\pm3.02}$ | $88.06_{\pm0.97}$ | $90.12_{\pm0.91}$ | $90.28_{\pm0.88}$ | $90.57_{\pm0.89}$ | $93.96_{\pm0.03}$ | 0.871M | 500.0× |
| Reddit (435.5M) | 0.5% | $69.92_{\pm2.32}$ | $78.68_{\pm0.94}$ | $60.14_{\pm1.84}$ | $91.14_{\pm0.59}$ | $91.06_{\pm0.93}$ | $91.73_{\pm0.52}$ | $91.84_{\pm0.73}$ | $93.96_{\pm0.03}$ | 2.178M | 200.0× |

## 4 EXPERIMENTS

In this section, we conduct experiments on six node classification graphs and three graph classification benchmarks to answer the following research questions:

- **RQ1.** How effective is our EXGC *w.r.t* efficiency and accuracy?
- **RQ2.** Can the design of EXGC be transferred to the state-of-the-art graph condensation frameworks (*e.g.,* DosGCond)?
- **RQ3.** What is the impact of the designs (*e.g.,* the backbone explainers) on the results? Is there a guideline for node selection?
- **RQ4.** Does the condensed graph exhibit strong cross-architecture capabilities?

### 4.1 Experimental Settings

**Datasets.** To evaluate the effectiveness of EXGC, we utilize six node classification benchmark graphs, including four transductive graphs, Cora [25], Citeseer [43], Ogbn-Arxiv and Ogbn-Product [21] and two inductive graphs, *i.e.,* Flickr [63] and Reddit [19]. For a fair comparison, we adopt the setup outlined in [24] and document the performance of various frameworks on the aforementioned datasets. Without loss of generality, we also select three graph classification datasets for evaluation: the Ogbg-molhiv molecular dataset [21], the TUDatasets (DD) [34] and one superpixel dataset CIFAR10 [10].

**Backbones.** In this paper, we employ a wide range of backbones to systematically validate the capabilities of EXGC. We choose one representative model, GCN [25], as our training model for the gradient matching process.

- To answer **RQ1**, we follow GCond to employ three coreset methods (*Random*, *Herding* [48] and *K-Center* [14]) and two data condensation models (DC-Graph) and GCond provided in [24]. Here we showcase the detailed settings in Table 2.

**Table 2: We compare the information utilized during the processes of condensation, training, and testing. Here, $A', X'$ represent the condensed graph and its features, while $A, X$ denote the original graph and its features, respectively.**

| | DC | DC-Graph | GCond-X (EXGC-X) | GCond (EXGC) |
|---|---|---|---|---|
| Condensation | $X_{train}$ | $X_{train}$ | $A_{train}, X_{train}$ | $A_{train}, X_{train}$ |
| Training | $X'$ | $X'$ | $X'$ | $A', X'$ |
| Test | $X_{test}$ | $A_{test}, X_{test}$ | $A_{test}, X_{test}$ | $A_{test}, X_{test}$ |

- To answer **RQ2**, we choose the current SOTA graph condensation method, DosGCond as backbone [23]. DosGCond eliminates the parameter optimization process within the inner loop of GCond, allowing for one-step optimization. This substantially reduces the time required for gradient matching. We employ DosGCond to further assess the generalizability of our algorithm.
- To answer **RQ3**, we select the explanation methods for node in $\mathcal{S}$ based on gradient magnitude (SA) [1], global mask (GSAT) [32], local mask (GNNExplainer) [57] as well as random selection, to evaluate the extensibility of backbone explainers.
- To answer **RQ4**, we choose currently popular backbones, such as APPNP [26], SGC [49] and GraphSAGE [19] to verify the transferability of our condensed graph (GCN as backbone). We also include MLP for validation.

**Measurement metric.** To ensure a fair comparison, we train our proposed method alongside the state-of-the-art approaches under identical settings, encompassing learning rate, optimizer, and so forth. Initially, we generate three condensed graphs, each developed using training methodologies with distinct random seeds. Subsequently, a GNN is trained on each of these graphs, with this training cycle repeated thrice (**Record the mean of the run time**). To gauge the information retention of the condensed graphs, we

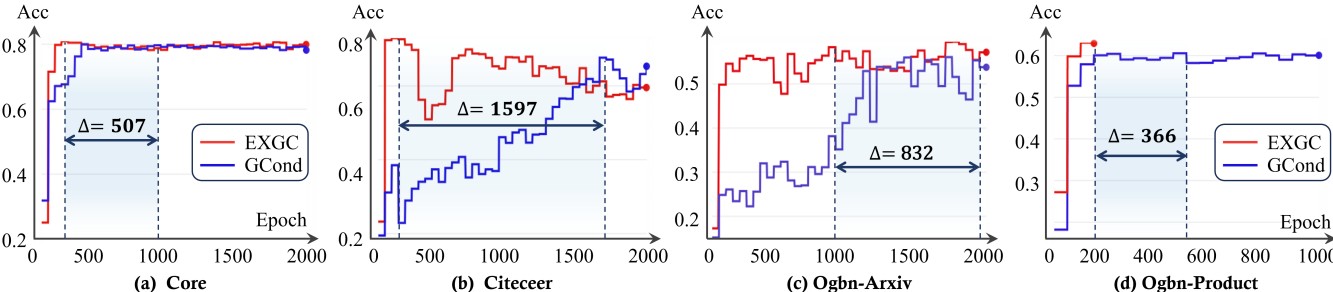

**Figure 3: The training process of EXGC and GCond across Cora, Citeseer, Ogbn-Arxiv and Ogbn-Product four benchmarks. We can observe that EXGC achieves optimal performance ahead by 507, 1097, 832, and 366 epochs respectively, at which points training can be terminated.**

proceed to train GNN classifiers, which are then tested on the real graph's nodes or entire graphs. By juxtaposing the performance metrics of models on these real graphs, we discern the informativeness and efficacy of the condensed graphs. All experiments are conducted in three runs, and we report the mean performance along with its variance.

## 4.2 Main Results (RQ1)

In this subsection, we evaluate the efficacy of a 2-layer GCN on the condensed graphs, juxtaposing the proposed methods, EXGC-X and EXGC, with established baselines. It's imperative to note that while most methods yield both structure and node features, note as $A'$ and $X'$, there are exceptions such as DC-Graph, GCond-X, and EXGC-X. Owing to the absence of structural output from DC-Graph, GCond-X, and EXGC-X, we employ an identity matrix as the adjacency matrix when training GNNs predicated solely on condensed features. Nevertheless, during the inference, we resort to the complete graph in a transductive setting or the test graph in an inductive setting to facilitate information propagation based on the pre-trained GNNs. Table 1 delineates performance across six benchmarks spanning various backbones, from which we can make the following observations:

**Obs 1. EXGC and EXGC-X consistently outperform other baselines** under extremely large condensation rates, thereby validating their exceptional performance. To illustrate, on smaller datasets such as Citeseer and Cora, our models achieve compression rates ranging from 0.3% ~0.4%, representing an improvement of approximately 3.84% to 8.15% over the current state-of-the-art model, GCond and GCond-X. On larger graphs like Ogbn-product and Reddit, EXGC surpasses GCond by 0.56% to 0.78% when aiming for a 0.5% compression rate. These findings underscore the substantial contributions of iterative optimization strategy of the subset of the nodes in $\mathcal{S}$ solely to the field of graph condensation (see Table 1). Additionally, our visualization results in Table 1 reveal that the graphs we condensed exhibit high density and compactness, with edges serving as efficient carriers of dense information, thereby facilitating effective information storage.

**Obs 2. Both EXGC and EXGC-X can achieve an extreme compression rate compared with the original graph** without significant performance degradation. On all six datasets, when compressing the original graph to a range of 0.05% to 5% of its original size, the compressed graph consistently maintains the performance

**Table 3: Comparing the time consumption and performance across different backbones. All results in seconds should be multiplied by 100. We activate 5% of the nodes every 50 epochs and stop training if the loss does not decrease for 4 consecutive epochs, subsequently reporting the results (results should be multiplied by 100).**

| Dataset | Ratio | GCond | EXGC | DosGCond | EXDos |
|---|---|---|---|---|---|
| Cora | 0.4% | 29.78s (72.76%) | 6.89s (81.13%) | 3.22s (74.05%) | 1.13s (81.64%) |
| Citeseer | 0.3% | 30.12s (63.98%) | 2.67s (67.45%) | 2.83s (67.73%) | 0.56s (69.81%) |
| Ogbn-arxiv | 0.05% | 184.90s (57.39%) | 96.31s (57.22%) | 20.49s (58.22%) | 5.60s (58.63%) |
| Flicker | 0.1% | 8.77s (46.81%) | 4.54s (47.21%) | 1.16s (46.04%) | 0.65s (46.80%) |
| Reddit | 0.1% | 53.04s (89.56%) | 19.83s (89.86%) | 5.75s (87.45%) | 1.71s (89.11%) |
| DD | 0.2% | – | – | 1.48s (72.65%) | 0.59s (72.90%) |
| CIFAR10 | 0.1% | – | – | 3.57s (30.41%) | 1.85s (29.88%) |
| Ogbg-molhiv | 0.01% | – | – | 0.49s (73.22%) | 0.31s (73.46%) |

of the original data while significantly accelerating the inference speed. This enhancement proves to be highly advantageous for information extraction and reasoning. Furthermore, as the storage requirement is reduced to a fraction of the original data's size (even better in some cases), this greatly facilitates the transmission of graph-type information in the web space, underscoring the exceptional contribution of our model to web systems.

## 4.3 Generalizability on DosGCond (RQ2)

To answer **RQ2**, we choose a gradient-based explainer as the backbone explainer. We transfer the SA into the current SOTA graph condensation method, DosGCond, and named as EXDos. We record the **performance** and **training time** of each backbone. As shown in Table 3 and Figure 3, we can make the observations as following:

**Obs 3. Upon incorporating the backbone explainers, significant reductions in the training time of GCond are achieved.** Furthermore, when our approach is applied to DosGCond, EXDos still exhibits substantial efficiency gains. Specifically, we observe efficiency improvements ranging from 1.78 ~ 5.05 times on five node classification datasets and speed enhancements of 1.58 ~ 2.51 times on graph classification datasets. These findings validate the effectiveness of our algorithm, offering a viable solution for efficient data compression.

**Obs 4. When employing the backbone explainers, the algorithm accelerates without noticeable performance decline.** As shown in Table 3, we find that on the eight datasets, the model consistently achieves acceleration without evident performance deterioration. Particularly, it gains performance improvements ranging from 2.08% ~ 9.26% on the Cora and Citeseer datasets. These

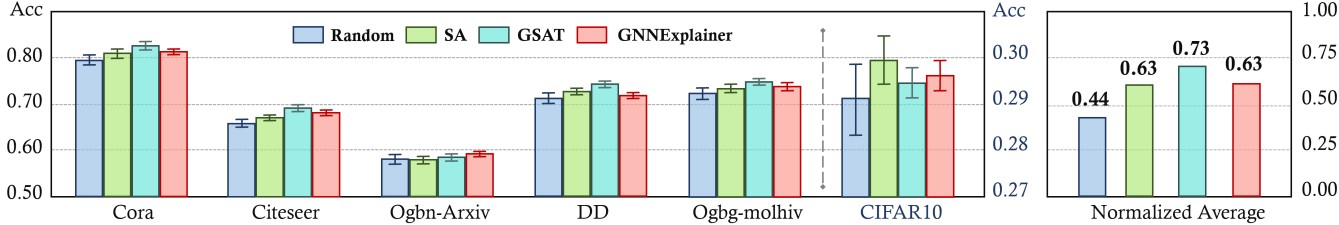

**Figure 4: Performance comparison across six benchmarks under various explanation methods.**

findings demonstrate that while reducing training and inference time, our approach does not lead to performance degradation and can even enhance the performance of the condensed graph.

**Table 4: Time comusing of different backbone explainers. We set the compress ratio of Cora, Citesser and Ogbn-Arxiv as 0.4%, 0.3%, 0.05%, respectively. As for graph classification, we set DD, CIFAR10 and Ogbg-molhiv as 0.2%, 0.1% and 0.01%. All displayed results should be multiplied by 100.**

| Method | EXGC | | | EXDO | | |
|---|---|---|---|---|---|---|
| | Cora | Citeseer | Ogbn-Arxiv | DD | CIFAR10 | Ogbg-molhiv |
| Random | 7.24s | 2.84s | 7.32s | 0.87s | 2.07s | 0.32s |
| SA | 6.89s | 2.67s | 6.31s | 0.59s | 1.85s | 0.31s |
| GSAT | 10.37s | 4.50s | 9.45s | 0.76s | 2.61s | 0.34s |
| GNNEXplainer | 11.62s | 5.97s | 11.23s | 0.99s | 2.80s | 0.42s |

## 4.4 Selection guidelines of EXGC (RQ3)

In this section, we choose a backbone explainer for nodes in $S$ based on gradient magnitude (SA), random selection, and explainable algorithms (GNNExplainer and GSAT). Our target is to determine whether different backbone explainers influence the performance of the EXGC. We employ a 2-layer GCN and introduce an early stopping mechanism, halting the network training if the loss fails to decrease over 4 consecutive epochs. Subsequently, we monitor and record the model's performance. By leveraging various backbone explainers, every 50 epochs, we select an additional 5% of the elements in X'. Here we can make the observations:

**Obs 5.** As shown in Table 4, EXDO denotes our backbone explainers transitioned into the overall framework of DosGCond. We discovered that the convergence time is similar for both random and gradient-based explainers. However, explainable algorithms necessitate considerable time due to the training required for the corresponding explainer. Intriguingly, despite the GNNExplainer-based selection method incurring the most substantial time cost, it still remains lower than the conventional DosGCond algorithm (see Table 3). Going beyond our explanation strategies, we need to further observe the performance of models under different algorithms to better assist users in making informed trade-offs.

**Obs 6.** After examining the efficiency, as illustrated in Figure 4, we observed that while balancing efficiency, GSAT can achieve the best results. In contrast, GNNExplainer has the lowest efficiency. Interestingly, while SA and random have similar efficiencies, SA manages to yield superior results in comparison.

**Table 5: Transferability of condensed graphs from the different architectures. Test performance across three popular GNN backbones, i.e., APPNP, SGC and GraphSAGE using 2-layer GCN as training setting is exhibited.**

| Method | GCond (backbone=GCN) | | | EXGC | | |
|---|---|---|---|---|---|---|
| | APPNP | SGC | SAGE | APPNP | SGC | SAGE |
| Cora | $69.32_{\pm4.26}$ | $67.95_{\pm6.10}$ | $60.34_{\pm4.83}$ | $75.17_{\pm3.93}$ | $74.02_{\pm4.88}$ | $66.49_{\pm4.25}$ |
| Citeseer | $61.27_{\pm5.80}$ | $62.43_{\pm4.52}$ | $61.74_{\pm5.01}$ | $67.34_{\pm3.83}$ | $68.58_{\pm4.42}$ | $66.62_{\pm4.17}$ |
| Ogbn-Arxiv | $58.50_{\pm1.66}$ | $59.11_{\pm1.35}$ | $59.04_{\pm1.13}$ | $59.37_{\pm0.89}$ | $60.07_{\pm1.82}$ | $58.72_{\pm0.99}$ |
| Flicker | $45.94_{\pm2.37}$ | $45.82_{\pm3.73}$ | $43.46_{\pm2.65}$ | $44.06_{\pm1.72}$ | $46.15_{\pm2.18}$ | $45.10_{\pm2.43}$ |
| Reddit | $85.42_{\pm1.76}$ | $87.33_{\pm2.97}$ | $84.80_{\pm1.34}$ | $87.46_{\pm2.73}$ | $86.10_{\pm1.55}$ | $87.59_{\pm2.92}$ |

## 4.5 Transferability of EXGC (RQ4)

Finally, we illustrate the transferability of condensed graphs from the different architectures. Concretely, we show test performance across different GNN backbones using a 2-layer GCN as the training setting. We employ popular backbones, APPNP, SGC and Graph-SAGE, as test architectures. Table 5 exhibits that:

**Obs 7.** Across five datasets, our algorithm consistently outperforms GCond and demonstrates relatively lower variance, validating the effectiveness of our approach. Notably, on the Cora dataset, our model achieves a performance boost of nearly 6.0%~7.0%. On the Citesser, we can observe that our framework achieves a performance improvement of approximately 5% to 6% over GCond. These results all underscore the transferability of our algorithm.

## 5 LIMITATION & CONCLUSION

**Limitation.** Our EXGC mitigates redundancy in the synthetic graph during training process without benefiting inference speed in downstream tasks. Moving forward, we aim to refine our algorithm to directly prune redundant nodes from the initialization of the synthetic graph, enabling simultaneous acceleration of both training and application phases. Additionally, we hope to adopt more advanced explainers in the future to better probe the performance boundaries.

**Conclusion.** In this work, we pinpoint two major reasons for the inefficiency of current graph condensation methods, i.e., the concurrent updating of a vast parameter set and the pronounced parameter redundancy. To address these limitations, we first employ the Mean-Field variational approximation for convergence acceleration and then incorporate the leading explanation techniques (e.g., GNNExplainer and GSAT) to select the important nodes in the training process. Based on these, we propose our EXGC, the efficient and explainable graph condensation method, which can markedly boost efficiency and inject explainability.

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

## A RELATED WORK

**Graph neural networks (GNNs).** GNNs [10, 19, 25, 55] handle variable-sized, permutation-invariant graphs and learn low-dimensional representations through an iterative process that involves transferring, transforming, and aggregating representations from topological neighbors. Though promising, GNNs encounter significant inefficiencies when scaled up to large or dense graphs [44]. To address this challenge, existing research lines prominently focus on **graph sampling** and **graph distillation** as focal points for enhancing computational efficiency.

**Graph Sampling & Distillation.** Graph sampling alleviates the computational demands of GNNs by selectively sampling sub-graphs or employing pruning techniques [5, 6, 11, 17, 27, 41]. Nevertheless, aggressive sampling strategies may precipitate significant information loss, potentially diminishing the representational efficacy of the sampled subset. In light of this, the research trajectory of graph distillation [37, 38, 58] is influenced by **d**ataset **d**istillation (DD), which endeavors to distill (compress) the embedded knowledge

within raw data into synthetic counterparts, ensuring that models trained on this synthetic data retain performance [4, 35, 46, 67]. Recently, within the domain of graph distillation, the notion of graph condensation [23, 24] via training gradient matching serves to compress the original graph into an informative and synthesized set, which also resides within the scope of our endeavor.

**Graph Lottery Ticket (GLT) Hypothesis.** The Lottery Ticket Hypothesis (LTH) articulates that a compact, efficacious subnetwork can be discerned from a densely connected network via an iterative pruning methodology [15, 16, 65]. Drawing inspiration from the concepts of LTH, [6] pioneered in amalgamating the concept of *graph samping* with *GNN pruning*, under the umbrella of Graph Lottery Ticket (GLT) research trajectory. Precisely, GLT is conceptualized as a coupling of pivotal core subgraphs and a sparse sub-network, which can be collaboratively extracted from the comprehensive graph and the primal GNN model. The ensuing amplification of GLT theory [59], coupled with the advent of novel algorithms [20, 29, 36, 47], has significantly enriched the graph pruning research narrative, delineating GLT as a prominent cornerstone in this field.

**Explainable Graph Learning.** Another line of research intimately related to our work delves into the explainability of graph learning. This line aims to reveal the black-box of the decision-making process by identifying salient subgraphs named *rationales*. Specifically, IB [53] and IB-subgraph [61] transplant the concept of information bottlenecks into graph learning to pinpoint compact yet informative subgraphs; GSAT [32] utilizes a stochastic attention mechanism to assign a probability to each edge in the input graph, determining whether it should be chosen as part of the explanatory subgraphs; DIR [54], based on causal inference, removes the label-irrelevant features from the original graph and treating the remaining portion as the graph's rationale; GREA [30] employs data augmentation methods based on both removal and replacement to target the original graph, aiming to explore the salient nodes.

## B DERIVATION OF THE EQUATION 4 AND 5

In this section we detail the derivation process of the Equation 4 and Equation 5 in Section 3.1. Specifically, to find a optimal model parameters $\Phi$, the objective can be formulated as:

$$\Phi = \arg\max_{\Phi} \log P(\nabla_\theta \mid \Phi). \tag{26}$$

We define the value of this logarithm as $F(\Phi)$ and rewrite it:

$$F(\Phi) = \log \sum_{\mathbf{X}'} P\left(\mathbf{X}'\right) \frac{P\left(\mathbf{X}', \nabla_\theta \mid \Phi\right)}{P\left(\mathbf{X}'\right)}, \tag{27}$$

then we can derive a low bound of $F(\Phi)$ according to the Jensen Inequality:

$$F(\Phi) \geq L(\Phi) = \sum_{\mathbf{X}'} P\left(\mathbf{X}'\right) \log \frac{P\left(\mathbf{X}', \nabla_\theta \mid \Phi\right)}{P\left(\mathbf{X}'\right)}, \tag{28}$$

where $L(\Phi)$ is the Variational Lower Bound of our objective.

To maximize the objective of $L(\Phi)$, we endeavour to derive the gap between $L(\Phi)$ and $F(\Phi)$ following:

$$
\begin{aligned}
L(\Phi) &= \sum_{\mathbf{X}'} P\left(\mathbf{X}'\right) \log \frac{P\left(\nabla_\theta, \mathbf{X}'|\Phi\right)}{P\left(\mathbf{X}'\right)} \\
&= \sum_{\mathbf{X}'} P\left(\mathbf{X}'\right) \log \frac{P\left(\mathbf{X}' \mid \nabla_\theta, \Phi\right) P\left(\nabla_\theta|\Phi\right)}{P\left(\mathbf{X}'\right)} \\
&= \log P\left(\nabla_\theta|\Phi\right) - \sum_{\mathbf{X}'} P\left(\mathbf{X}'\right) \ln \frac{P\left(\mathbf{X}'\right)}{P\left(\mathbf{X}' \mid \nabla_\theta, \Phi\right)} \\
&= F(\Phi) - KL(P(\mathbf{X}')\|p(\mathbf{X}' \mid \nabla_\theta; \Phi)).
\end{aligned}
\tag{29}
$$

It is well known that the KL divergence is non-negative. Therefore, with $\Phi$ fixed and optimizing $\mathbf{X}'$, maximizing this lower bound is equivalent to:

$$
P(\mathbf{X}') \leftarrow P(\mathbf{X}'|\nabla_\theta, \Phi).
\tag{30}
$$

Moreover, maximizing $L(\Phi)$ is equivalent to maximizing the ELBO:

$$
\text{ELBO} \rightarrow E_{\mathbf{X}'|\nabla_\theta, \Phi}[\log \frac{P(\mathbf{X}', \nabla_\theta \mid \Phi)}{P(\mathbf{X}' \mid \nabla_\theta, \Phi)}]
\tag{31}
$$

by maximizing the conditional probability expectation following:

$$
\begin{aligned}
\Phi &= \arg\max_\Phi L(\Phi) \\
&= \arg\max_\Phi \sum_{\mathbf{X}'} p\left(\mathbf{X}' \mid \nabla_\theta, \Phi\right) \ln \left\{ \frac{p(\nabla_\theta, \mathbf{X}' \mid \Phi)}{p\left(\mathbf{X}' \mid \nabla_\theta, \Phi\right)} \right\} \\
&= \arg\max_\Phi (\sum_{\mathbf{X}'} p\left(\mathbf{X}' \mid \nabla_\theta, \Phi\right) \ln p(\nabla_\theta, \mathbf{X}' \mid \Phi) - \\
&\quad \underbrace{\sum_{\mathbf{X}'} p\left(\mathbf{X}' \mid \nabla_\theta, \Phi\right) \ln p\left(\mathbf{X}' \mid \nabla_\theta, \Phi\right)}_{const}),
\end{aligned}
\tag{32}
$$

which is exactly what the current M-step does.

## C  DERIVATION OF THE MF APPROXIMATION

Let's start with Equation 7 to derive the optimized E-step based on mean-filed approximation. Specifically, by substituting Equation 6 into the ELBO in Equation 4 we obtain:

$$
\begin{aligned}
\text{ELBO} &= \int \prod_{i=1}^{N} P\left(x_i'\right) \log P(\nabla_\theta, \mathbf{X}') d\mathbf{X}' \\
&\quad - \int \prod_{i=1}^{N} P\left(x_i'\right) \log \prod_{i=1}^{N} P\left(x_i'\right) d\mathbf{X}'.
\end{aligned}
\tag{33}
$$

For simplicity, let's make some variable assumptions below:

$$
\begin{aligned}
\mathcal{A} &= \prod_{i=1}^{N'} P\left(x_i'\right) \log P(\mathbf{X}', \nabla_\theta) d\mathbf{X}' \\
\mathcal{B} &= \int \prod_{i=1}^{N'} P\left(x_i'\right) \log \prod_{i=1}^{N'} P\left(x_i'\right) d\mathbf{X}'.
\end{aligned}
\tag{34}
$$

In this case, the ELBO can be rewritten as:

$$
\text{ELBO} = \mathcal{A} - \mathcal{B}.
\tag{35}
$$

Before deriving $\mathcal{A}$ and $\mathcal{B}$, we first fix the complementary set of $x_j'$, i.e., $\mathbf{X}'_{\setminus j} = \{x_1', \ldots x_{j-1}', x_{j+1}', \ldots, x_{N'}'\}$. Then $\mathcal{A}$ is equal to:

$$
\mathcal{A} = \int P\left(x_j'\right) \int \prod_{i=1, i \neq j}^{N'} P\left(x_i'\right) \log P(\nabla_\theta, \mathbf{X}') d_{i \neq j} x_i' dx_j',
\tag{36}
$$

where:

$$
\int \prod_{i=1, i \neq j}^{N'} P\left(x_i'\right) \log P(\mathbf{X}', \nabla_\theta) d_{i \neq j} x_i' = E_{\prod_{i=1, i \neq j}^{N'} P(x_i')}\left[\log P(\mathbf{X}', \nabla_\theta)\right].
\tag{37}
$$

By substituting Equation 37 into the Equation 36 we obtain:

$$
\mathcal{A} = \int P\left(x_j'\right) \cdot E_{\prod_{i=1, i \neq j}^{N'} P(x_i')}\left[\log P(\mathbf{X}', \nabla_\theta)\right] dx_j'.
\tag{38}
$$

Next we focus on the term $\mathcal{B}$. We first rewritten it following:

$$
\mathcal{B} = \int \prod_{i=1}^{N'} P\left(x_i'\right) \cdot \left[\log P\left(x_1'\right) + \log P\left(x_2'\right) + \cdots + \log P\left(x_{N'}'\right)\right] d\mathbf{X}'.
\tag{39}
$$

Note that for each terms in $\mathcal{B}$ we have:

$$
\int \prod_{i=1}^{N'} P\left(x_i'\right) \cdot \log P\left(x_1'\right) = \int P\left(x_1'\right) \log P\left(x_1'\right) dx_1'.
\tag{40}
$$

Hence, the value of $\mathcal{B}$ can be simplified as:

$$
\mathcal{B} = \sum_{i=1}^{N'} \int P\left(x_i'\right) \log P\left(x_i'\right) dx_i'.
\tag{41}
$$

Since $\mathbf{X}_{\setminus j}$ is fixed, we can separate out the constants $C$ from $\mathcal{B}$:

$$
\mathcal{B} = \int P\left(x_j'\right) \log P\left(x_j'\right) dx_j' + C.
\tag{42}
$$

Combining the expression for $\mathcal{A}$ mentioned above, we can obtain a new form of ELBO:

$$
\begin{aligned}
\text{ELBO} &= \mathcal{A} - \mathcal{B} = \int P\left(x_j'\right) \log \frac{E_{\prod_{i=1, i \neq j}^{N'} P(x_i')}\left[\log P(\mathbf{X}', \nabla_\theta)\right]}{P\left(x_j'\right)} dx_j' \\
&= -KL\left(P(x_j')\| \log E_{\prod_{i=1, i \neq j}^{N'} P(x_i')}\left[\log P(\mathbf{X}', \nabla_\theta)\right]\right) \leq 0.
\end{aligned}
\tag{43}
$$

Therefore, when KL is equal to 0, ELBO can reach its maximum value, so the value of $P\left(x_j'\right)$ derived here is:

$$
P\left(x_j'\right) = E_{\prod_{i=1, i \neq j}^{N'} P(x_i')}\left[\log P(\mathbf{X}', \nabla_\theta)\right].
\tag{44}
$$

The methods for solving for distributions of $P\left(x_i'\right)$ for $i \in \{1, \ldots, j-1, j+1, \ldots, N'\}$ are the same.

## D  DERIVATION OF THE GDIB

In this section we focus on the detailed derivation process in Section 3.4, which mainly contribute to the instantiation process of GDIB:

$$
\arg\max_{\mathcal{S}_{sub}} I\left(\mathcal{S}_{sub}; \nabla_\theta'\right) - \beta I\left(\mathcal{S}_{sub}; \mathcal{S}\right), \text{ s.t. } \mathcal{S}_{sub} \in \mathbb{G}_{sub}(\mathcal{S}).
\tag{45}
$$

At first, for the first term in GDIB, *i.e.*, $I\left(\mathcal{S}_{sub}; \nabla_\theta'\right)$, by definition:

$$I(\mathcal{S}_{sub}; \nabla_\theta') = H(\nabla_\theta') - H(\nabla_\theta' \mid \mathcal{S}_{sub})$$

$$= E_{\nabla_\theta', \mathcal{S}_{sub}}\left[\log \frac{P\left(\mathcal{S}_{sub} \mid \nabla_\theta'\right)}{P\left(\nabla_\theta'\right)}\right]. \tag{46}$$

Since $P\left(\mathcal{S}_{sub} \mid \nabla_\theta'\right)$ is intractable, an variational approximation $Q\left(\mathcal{S}_{sub} \mid \nabla_\theta'\right)$ is introduced for it. Then the LBO of the $I(\nabla_\theta', \mathcal{S}_{sub})$ can be obtained following:

$$\begin{aligned}
I\left(\mathcal{S}_{sub}; \nabla_\theta'\right) =& E_{\mathcal{S}_{sub}, \nabla_\theta'}\left[\log \frac{Q\left(\nabla_\theta' \mid \mathcal{S}_{sub}\right)}{P(\nabla_\theta')}\right] \\
&+ E_{\mathcal{S}_{sub}, \nabla_\theta'}\left[\log \frac{P\left(\nabla_\theta' \mid \mathcal{S}_{sub}\right)}{Q\left(\nabla_\theta' \mid \mathcal{S}_{sub}\right)}\right] \\
=& E_{\mathcal{S}_{sub}}\left[\text{KL}\left(Q\left(\nabla_\theta' | \mathcal{S}_{sub}\right) \| P\left(\nabla_\theta'\right)\right)\right] \\
&+ E_{\mathcal{S}_{sub}}[\text{KL}\left(P\left(\nabla_\theta' | \mathcal{S}_{sub}\right) \| Q\left(\nabla_\theta' | \mathcal{S}_{sub}\right)\right)] \\
\geq& \underbrace{E_{\mathcal{S}_{sub}}\left[\text{KL}\left(Q\left(\nabla_\theta' | \mathcal{S}_{sub}\right) \| P\left(\nabla_\theta'\right)\right)\right]}_{\text{LBO}}.
\end{aligned} \tag{47}$$

Then, for the second term in GDIB, *i.e.*, $I\left(\mathcal{S}_{sub}; \nabla_\theta'\right)$, by definition:

$$I(\mathcal{S}, \mathcal{S}_{sub}) = H(\mathcal{S}) - H(\mathcal{S} \mid \mathcal{S}_{sub})$$

$$= E_{\mathcal{S}, \mathcal{S}_{sub}}\left[\log \frac{P\left(\mathcal{S}_{sub} \mid \mathcal{S}\right)}{P\left(\mathcal{S}_{sub}\right)}\right]. \tag{48}$$

Considering that $P(\mathcal{S}_{sub})$ is intractable, an variational approximation $R(\mathcal{S}_{sub})$ is introduced for the marginal distribution $P(\mathcal{S}_{sub}) = \sum_{\mathcal{S}} P\left(\mathcal{S}_{sub} \mid \mathcal{S}\right) P(\mathcal{S})$. Then the UBO of the $I(\mathcal{S}, \mathcal{S}_{sub})$ can be obtained following:

$$\begin{aligned}
I(\mathcal{S}, \mathcal{S}_{sub}) =& E_{\mathcal{S}_{sub}, \mathcal{S}}\left[\log \frac{P\left(\mathcal{S}_{sub} \mid \mathcal{S}\right)}{R\left(\mathcal{S}_{sub}\right)}\right] - \text{KL}\left(P\left(\mathcal{S}_{sub}\right) \| R\left(\mathcal{S}_{sub}\right)\right) \\
\leq& \underbrace{E_{\mathcal{S}}\left[\text{KL}\left(P\left(\mathcal{S}_{sub} \mid \mathcal{S}\right) \| R\left(\mathcal{S}_{sub}\right)\right)\right]}_{\text{UBO}}.
\end{aligned}$$

$$\tag{49}$$

The main paper presents an instantiation of $P\left(\mathcal{S}_{sub} | \mathcal{S}\right)$ which assigns the importance score $p_i$ (*i.e.*, the probability of being selected into $\mathcal{S}_{sub}$) to the $i$-th node in $\mathcal{S}$. Additionally, the distribution $R$ is specified as a Bernoulli distribution with parameter $r$ (*i.e.*, each node is selected with probability $r$). This instantiation is consistent with the information constraint $\ell_I$ proposed by GSAT [32], where $r$ falls within the range of $(0, 1)$, resulting in a collapse of the UBO to the $\ell_I$:

$$\ell_I = \sum_{i \in 1,2,\ldots,N} p_i \log \frac{p_i}{r} + (1 - p_i) \log \frac{1 - p_i}{1 - r}. \tag{50}$$

When $r \to 0$ we have:

$$p_i \log \frac{p_i}{r} \gg (1 - p_i) \log \frac{1 - p_i}{1 - r}. \tag{51}$$

Then the Equation 50 collapses to:

$$\ell_I = \sum_{i \in 1}^{N'} p_i \log \frac{p_i}{r}. \tag{52}$$

Since the value of Equation 52 is proportional to the value of $p_i$, when $r \to 0$, $\ell_I$ can be instantiated as the $l_1$-norm of $p_i$.