# OpenReview forum: "EXGC: Bridging Efficiency and Explainability in Graph Condensation"
_ACM.org/TheWebConf/2024/Conference — TheWebConf24_

### Official Review · Reviewer_r2xF · 2023-11-19

**Novelty:** 5
**Technical Quality:** 5

**Review:**

This paper studied the problem of graph condensation, and proposed using mean-field approximation to boost the computation of probabilities in the EM procedure. Furthermore, to resolve the issue of node redundancy in previous methods, the authors used explanation strategies to filter out less informative nodes to further decrease the size of the final condensed graph. Extensive simulation results are included in the paper, and the performance improvements on some of the datasets seem promising.

In general, I think the problem considered here is fairly interesting, and the proposed method is performing reasonably well even with multiple steps of approximations. I believe that the technical content presented in this paper is sufficient for publication, and the paper itself is well-structured. Nevertheless, I would still recommend the authors make full use of the unlimited appendix to expand the proofs, as some of the notations in the proofs are not very technical. For example, in Eq. 30, what does the left arrow really mean?

**Questions:**

What does the suffix "-X" mean for the method "GCond-X" and "EXGC-X"? I do not think this is explained in the main text.

**Reviewer Confidence:**

2: The reviewer is willing to defend the evaluation, but it is likely that the reviewer did not understand parts of the paper

**Scope:**

3: The work is somewhat relevant to the Web and to the track, and is of narrow interest to a sub-community

---

### Official Review · Reviewer_KUA3 · 2023-11-23

**Novelty:** 4
**Technical Quality:** 4

**Review:**

1. Summary: This paper presents a new method named EXGC to address two major inefficiencies in graph condensation (GCond): the concurrent updating of a vast parameter set and parameter redundancy. These inefficiencies are pronounced in large-scale web data graphs. The authors propose to use Mean-Field (MF) variational approximation to accelerate convergence and introduce the Gradient Information Bottleneck (GDIB) principle to prune redundancy. By integrating leading explanation techniques such as GNNExplainer and GSAT, EXGC not only boosts efficiency but also adds an element of explainability to the graph condensation process.

2. Strength:
(1) This paper aims to bridge the goals of efficiency and explainability in GCond, presenting a trial that is both innovative and highly relevant in today's data-driven landscape. Especially when considering pruning the redundancy, the proposed model applies the existing explainable models to filter the informative nodes. This operation becomes the key where EXGC can achieve a high training efficiency and storage efficiency.
(2) The explored problem in the paper is well motivated, reflecting a deep understanding of the current challenges and needs in the field of graph representation learning. The authors have identified a critical gap in the existing methodologies, particularly in how they handle large-scale graph data with regard to efficiency and explainability. This motivation is rooted in the practical difficulties faced by practitioners who deal with expansive web data graphs, where the computational and storage overheads are significant barriers.
(3) The experimental results show a significant improvement based on the performance of existing methods. The authors have meticulously compared their method against a range of established benchmarks, showcasing substantial leaps in multiple performance metrics.

3. Weakness:
(1) The technique details are hard to follow. This paper includes too many details of variational approximation but is in short of high-level ideas to help reviewers better understand the intuition of the multiple modules like MGCond and GDIB. If these details are summarized into several important lemmas and more high-level demonstration is provided, the readability can be further enhanced.
(2) This paper clearly lists two significant limitations in existing methods, but lacks further explanation of how these limitations exist. For example, this paper mentions multiple times that ” the number of to-be-updated parameters in E-step linearly increases with the number of nodes”, but there is no explanation of how this situation holds. Maybe it can be demonstrated using the theoretical references or experimental results.
(3) It is not very clear why EXGC is better than other GCond methods. This paper discusses a lot about the design of the model itself, but lacks the discussion of how EXGC exceeds other SOTA GCond methods. This kind of discussion is short of objectivity and not friendly to the readers who are not specifically in the GCond area. Moreover, it seems that this paper only compares the storage with the original graph, and a comparison with other GCond methods should be provided.
(4) The Unified Paradigm of GCond in Section 3.1 seems to be an existing framework. Therefore it should be put into the background.
(5) The experimental part can be further enhanced by adding more experiments about other GCond baselines.

**Questions:**

(1) Is this paper the first work to utilize the explainable models in GCond (for filtering the nodes)? If yes, this paper can further highlight this point. If not, this paper should add more discussion of novelty in Section 3.4.
(2) Can you provide more results when compared with other GCond methods? For example, we can see that Ogbn-Product datasets can be reduced from 1412.5M to 7.063M. However, the performance of other competitive GCond baselines is unknown.

**Reviewer Confidence:**

2: The reviewer is willing to defend the evaluation, but it is likely that the reviewer did not understand parts of the paper

**Scope:**

4: The work is relevant to the Web and to the track, and is of broad interest to the community

---

### Official Review · Reviewer_xrwK · 2023-11-25

**Novelty:** 5
**Technical Quality:** 4

**Review:**

The authors propose two ideas for accelerating graph condensation. The first idea is to apply mean-field approximation to parameter updates, and the second idea is to focus on updating a subset of parameters, rather than updating all parameters. The subset is chosen using GNN explainers. The experimental results show that the proposed ideas improve condensation efficiency without compromising performance.

S1. The paper is well-written and easy to follow.

S2. It highlights a significant concern in graph condensation and provides interesting observations that serve as the foundation for the proposed methods.

S3. The proposed method has a solid theoretical foundation.

S4. The experiments are comprehensive, demonstrating the advantages of the proposed methods in various aspects.

S5. The authors recognized important limitations in their approach.

W1. Although the paper claims that the proposed method is explainable, it lacks empirical evidence to support this assertion. It appears that the authors make this claim primarily due to the incorporation of GNN explainers into their method.

W2. The presentation of experimental results is somewhat confusing. Firstly, different experiments employ varying backbones. I suggest using DosCond as the primary backbone, as it consistently outperforms GCond in all aspects. Second, assigning new names (e.g., EXDos, EXDO) to variants also leads to confusion. Third, it is unconventional to request readers to multiply the presented results by 100.

W3. My main concern is that the proposed method's advantage over DosGCond, a state-of-the-art technique, appears limited, as shown in Table 3. Especially in terms of condensation performance, the improvement seems minor. Additionally, the assertion of a 2-4 times speed increase raises questions because the paper lacks details about the specific implementations, which largely affect the speed.

W4. Although the proposed method is theoretically grounded, the improvement it brings is not theoretically established, such as through a complexity analysis and convergence time analysis.

**Questions:**

See W1, W3, and W4.

**Reviewer Confidence:**

3: The reviewer is confident but not certain that the evaluation is correct

**Scope:**

3: The work is somewhat relevant to the Web and to the track, and is of narrow interest to a sub-community

---

### Official Review · Reviewer_gvtH · 2023-12-01

**Novelty:** 5
**Technical Quality:** 6

**Review:**

The authors tackle the limitations of current graph condensation approaches for supporting graph representation learning on large-scale graphs. As such, they propose the novel Efficient and eXplainable Graph Condensation, EXGC, method. This leverages convergence acceleration and redundancy pruning mechanisms and is experimentally evaluated on six node classification graphs and three graph classification benchmarks against state-of-the-art baselines.

+ The paper tackles an important problem through a series of interesting and original techniques.

+ The paper is technically rigurous and contains a detailed presentation of the proposed methodology.

+ The experimental analysis is extensive.

- The presentation of the paper could be further improved, by including a notation table for the technical section and by illustrating the main ideas on clarifying examples.

**Questions:**

Q1. What is the computational complexity of the EXGC method?

Q2. What are the reasons due to which EXGC is largely outperformed by the state-of-the-art DosGCond method across the datasets presented in Table 3?

Q3. How do topological characteristics of the considered graph datasets, e.g., density, degree distribution, affect the performance of the proposed approach across the different experimental scenarios the authors consider?

**Ethics Review Description:**

Nothing to report.

**Reviewer Confidence:**

2: The reviewer is willing to defend the evaluation, but it is likely that the reviewer did not understand parts of the paper

**Scope:**

4: The work is relevant to the Web and to the track, and is of broad interest to the community

---

### Decision · Program_Chairs · 2024-01-22

**Decision:**

Accept

**Comment:**

The reviewers agree that this is a good paper for the conference, but also give additional details on experiments / proofs by which the paper can be improved.